# Research on the High Temperature and High Pressure Gold-Plated Fiber Grating Dual-Parameter Sensing Measurement System

**DOI:** 10.3390/mi13020195

**Published:** 2022-01-27

**Authors:** Na Zhao, Zhongkai Zhang, Qijing Lin, Kun Yao, Liangquan Zhu, Yi Chen, Libo Zhao, Bian Tian, Ping Yang, Zhuangde Jiang

**Affiliations:** 1State Key Laboratory for Manufacturing Systems Engineering, Xi’an Jiaotong University, Xi’an 710049, China; zn2020@xjtu.edu.cn (N.Z.); yao_kun@outlook.com (K.Y.); zhuliangquan@stu.xjtu.edu.cn (L.Z.); libozhao@xjtu.edu.cn (L.Z.); t.b12@mail.xjtu.edu.cn (B.T.); ipe@xjtu.edu.cn (P.Y.); zdjiang@xjtu.edu.cn (Z.J.); 2Collaborative Innovation Center of High-End Manufacturing Equipment, Xi’an Jiaotong University, Xi’an 710054, China; 3Aeronautics and Astronautics Engineering College, Air Force Engineering University, Xi’an 710038, China; chenyikgd@163.com

**Keywords:** optical fiber sensor, gold-plated fiber Bragg grating, pressure, temperature sensing

## Abstract

In electrohydrostatic drive actuators, there is a demand for temperature and pressure monitoring in complex environments. Fiber Bragg grating (FBG) has become a promising sensor for measuring temperature and pressure. However, there is a cross-sensitivity between temperature and pressure. A gold-plated FBG is proposed and manufactured, and an FBG is used as a reference grating to form a parallel all-fiber sensing system, which can realize the simultaneous measurement of pressure and temperature. Based on the simulation software, the mechanical distribution of the pressure diaphragm is analyzed, and the fixation scheme of the sensor is determined. Using the demodulator to monitor the changes in the reflectance spectrum in real-time, the pressure and ambient temperature applied to the sensor are measured. The experimental results show that the temperature sensitivity of gold-plated FBG is 3 times that of quartz FBG, which can effectively distinguish the temperature changes. The pressure response sensitivity of gold-plated FBG is 0.3 nm/MPa, which is same as the quartz FBG. Through the sensitivity matrix equation, the temperature and pressure dual-parameter sensing measurement is realized. The accuracy of the temperature and pressure measurement is 97.7% and 99.0%, and the corresponding response rates are 2.7 ms/°C and 2 ms/MPa, respectively. The sensor has a simple structure and high sensitivity, and it is promising to be applied in health monitoring in complex environments with a high temperature and high pressure.

## 1. Introduction

The electro-hydrostatic actuator (EHA) is a highly integrated actuator [1,2], which can be widely used in the field of hydraulic transmission and control. However, the EHA structural system is complex and the working environment is harsh. There are defects, such as poor control accuracy, the difficulty in predicting failures, and low maintainability, which limit its operational reliability on major equipment. Monitoring the important working parameters of the temperature and pressure of the whole machine in the EHA system through sensing technology to form a feedback and regulation mechanism, can improve the control accuracy and reliability of the EHA.

At present, the reported electrical pressure and temperature integrated sensors are mainly used for normal temperature and low pressure monitoring [3,4,5], and there are few reports on integrated sensors that can be used in high temperature and high pressure harsh environments. Compared with the conventional electrical sensors, optical fiber sensors [6,7] have attracted attention for their stable chemical properties, anti-electromagnetic interference, compact structure, low cost, diverse functions, good insulation performance, and light weight. However, the research of optical fiber sensors mostly focuses on single-parameter measurement. To date, the optical fiber multi-parameter integrated sensors that meet the special needs of extreme environments are still problems that need to be tackled.

The related research of optical fiber temperature sensing mainly focuses on, for instance, the Mach–Zehnder interferometer [8,9,10], Michelson interferometer [11,12], Fabry–Perot interferometer [13,14], long period grating [15,16], and fiber Bragg grating (FBG) [17,18,19,20,21,22,23,24,25,26,27,28,29,30,31,32,33,34,35,36,37]. Although the above-mentioned various sensors can be directly used for temperature measurement, there are some shortcomings. For example, the structure of the Mach–Zehnder interferometer cannot be made into a probe structure; the structure of the Fabry–Perot interferometer is poor in stability; and the large measuring point size of the long-period grating leads to inaccurate measurements. In contrast, FBG has attracted much attention due to its small size, low cost, easy manufacturing, and high maturity. In 2003, NASA [17] adopted a distributed quartz fiber grating sensing system and installed the fiber sensing network on the X-38 space shuttle to realize the real-time monitoring of the temperature of the composite fuel tank. To date, the FBG sensor based on quartz fiber formed a relatively stable industrialized temperature detection system, with a temperature response sensitivity of about 0.01 nm/°C [18,19,20]. In 2021, Keith M.Alcock et al. [21] used the optical fiber sensor to measure the temperature of a lithium-ion battery, realizing the miniaturized installation of the optical fiber sensor. In the same year, Angela Brindisi et al. [22] equipped a fiber optic sensor on a small landing gear, based on the mechanical sensing performance of FBG, to evaluate whether there was a hard landing and the degree of the hard landing. In order to improve the sensitivity, people have made various attempts, including corrosion, coating, and other methods. Finally, it was found that corrosion does not help much to improve the temperature sensitivity of the grating, and it is easy to introduce physical interference, such as the refractive index and humidity. Therefore, optical fiber coating has become the focus of attention. Additionally, gold-plated optical fiber has the advantages of high expansion coefficients, good adhesion of gold atoms to optical fibers, and a mature manufacturing industry. Compared with quartz FBG, gold-plated FBG has a higher temperature response sensitivity, and gold-plated FBG also responds to strain, which can realize high temperature and high pressure measurements. In 2015, Monaghan et al. [23] used a metalized packaging method to improve the temperature sensitivity of FBG. In 2017, Liu Yanchao et al. [24] proposed a method for the in situ detection of lithium-ion batteries by pasting gold-plated fiber Bragg grating (FBG) sensors during the production of lithium-ion batteries. In 2019, Wu Hao et al. [25] used gold-plated grating and quartz grating cascades to realize the temperature and strain sensing measurements; the temperature response sensitivity was 26.5 pm/°C in the range of 30–70 °C, and the strain response sensitivity was 1.19 pm/µε up to 400 µε. In the same year, I. Laarossi et al. [26] used gold-plated gratings to measure the temperature and strain; the measurement response sensitivity was 1.10 pm/με and 3.7 pm/°C, respectively. In 2021, Yanjun Zhang et al. [27] conducted research on gold-plated FBG; the sensitization characteristics of the sensor were theoretically analyzed, and the response characteristics of the sensor were studied. To date, gold-plated FBG is mostly used in plasmon resonance [28,29,30] and battery biomarkers [31,32], and there have been few studies on the temperature and pressure.

Pressure value monitoring in EHA health monitoring is also very important. The pressure range of the pure optical fiber sensing structure is generally in the order of kPa [33,34,35,36], and with the help of a cantilever beam and diaphragm structure, it can reach the order of MPa [37,38,39,40,41]. In 2013, Lijun Li et al. [37] developed the FBG pressure sensor in order to meet the needs of coal mine production safety, with a sensitivity of 0.5983 nm/MPa in the range of 0 to 7.15 MPa. In 2016, Wang Hui et al. [38] designed a fiber grating pressure sensor combined with a cantilever beam, and the pressure sensitivity reached 3 × 103 nW/MPa in the measurement range of 0–6 MPa. In the same year, Hongtao Zhang [39] proposed a high-sensitivity pressure sensor based on FBG wavelength detection to measure the downhole pressure in oil and gas wells, and the sensitivity of the sensor can reach 230.9 pm/MPa in the range of 0 to 20 MPa. In 2017, Yiping Wang et al. [40] developed a high-sensitivity pressure sensor using phase shifted FBGs, and achieved a high-sensitivity measurement of 418.8 MHz/MPa in the sensing range of 0–4 MPa. In 2019, Zhen’an Jia et al. [41] proposed an FBG pressure sensor using a composite structure; the composite structure included a square diaphragm, a steel truss, and a vertical beam, and the pressure sensitivity of the sensor was 622.71 pm/MPa in the range of 0 to 2 MPa.

The above studies mostly focused on the measurement of the temperature or pressure, but the temperature and pressure responses of the sensor were all wavelength types. The combined effect of the temperature and pressure causes cross-interference between the data. How to overcome the multi-parameter interference and meet the needs of temperature and pressure sensor monitoring in a complex environment is the key point of later research. In order to achieve the simultaneous response to pressure and temperature, Nan Wang [42] of the PLA Naval Armament Department integrated and multiplexed the optical fiber pressure sensors and temperature compensation gratings to achieve rapid temperature compensation at low and normal temperatures, and complete 8 MPa large-scale high-precision pressure sensing; the response sensitivity was 0.15 nm/MPa. In 2019, Wenhua Wang et al. [43] proposed a Fabry–Perot interferometer and FBG cascaded fiber optic pressure sensor, and measured a pressure response of 0–1 MPa and a temperature response from 5.6 to 26.4 °C. In 2021, Qinggeng Fan et al. [44] designed a high-sensitivity square diaphragm pressure sensor based on FBG, and conducted theoretical and experimental verifications. The experimental results show that the pressure sensitivity of the sensor is 3.402 pm/kPa, in the range of 0–200 kPa, and the temperature response sensitivity is 19.29 pm/°C at 20–55 °C; this structure is suitable for low pressure measurement. In the existing papers, the research mostly focuses on the low-pressure and low-temperature section; however, the single-parameter measurement of temperature can already reach a high temperature [45]. However, for the temperature measurement under high pressure, the current research generally stays at room temperature, and there is less measurement and monitoring in the high-temperature and high-pressure complex environments.

In this paper, a high-temperature-resistant pressure diaphragm-type FBG temperature and pressure dual-parameter sensor is developed. The mechanical characteristics of the pressure sensitive diaphragm are simulated, the position range of the sensor on the pressure diaphragm is guided, and the working principle of the sensor is discussed. Based on the temperature and pressure characteristics of the gold-plated grating, which is different from the quartz grating, combined with the sensitivity matrix equation, the purpose of constructing a temperature and pressure dual-parameter sensor monitoring on the sensing probe is realized. The experimental results show that the optical fiber sensor has great potential in the simultaneous sensing and measurement of temperature and pressure.

## 2. Principle and Design

The preparation of gold-plated FBG is based on the photosensitive characteristics of optical fibers to produce FBG. By using ultraviolet light, some specific optical waveguide structures can be written into the optical fiber. After forming the optical fiber optical waveguide device using electron beam evaporation or the magnetron sputtering system gold plating method, the outer surface of the FBG can be gold-plated. FBG is packaged with a material with a large thermal expansion coefficient to improve the temperature sensitivity. The experimental design of the temperature and pressure sensing probe structure design is shown in Figure 1a. The overall diameter of the probe is 11 mm and the height is 5 mm. The sensing probe is divided into three parts, including a hexagonal nut-type cap for fixing the elastic diaphragm and the threaded connection structure below; the diaphragm is used to sense the pressure and the grating is pasted on it. The connection with the pressure supply device is based on a threaded connection end, which cooperates with a hexagonal nut to form a fixation to the diaphragm, and the physical diagram of the developed sensing probe is shown in Figure 1b,c. When the fluid pressure acts on the circular diaphragm, the diaphragm is deformed. The slight stretching and deformation of the fiber grating will cause the center wavelength of the fiber grating to shift, and the center wavelength shift will reflect the fluid pressure value. The gold-coated FBG and reference quartz FBG used in the experiments have the different grating pitches of 523.52 nm and 528.40 nm, respectively. Different grating pitches ensure that they have different center wavelengths, which is convenient for data analysis. In addition, they are all single-mode fibers with a fiber diameter of 250 µm and a FBG length of 300 mm. Considering the two-level difference of the grating length and diameter, the parallel connection of the gratings is selected for sensor installation, as shown in Figure 1d. The center wavelength of the reflection spectrum of the dual FBG is different, and the two sensing probes can be analyzed separately and the reflection spectrum can be coupled through the fiber coupler.

Combined with the working environment of the pressure sensor, and based on ANSYS software, a mechanical simulation of a circular pressure diaphragm with a diameter of 1.5 cm, a thickness of 2 mm, and a material of 0Cr17Ni12Mo2 (AISI316) are carried out. As shown in Figure 2, the stress distribution in the linear part of the diaphragm at the middle diameter of 1 cm is relatively uniform, so that the force part of the FBG is concentrated on the center of the diaphragm as much as possible to obtain a more uniform stress distribution. When fluid pressure acts on the circular diaphragm, the diaphragm is deformed, and the slight stretching and deformation of the FBG will cause the center wavelength of the fiber grating to shift, and the center wavelength shift will reflect the fluid pressure value.

The sensor measurement of the temperature and pressure environments can be obtained by designing a dual grating parallel structure with different response sensitivities. In terms of the temperature measurement, for the FBG structure, the thermo-optical effect and thermal expansion effect affect the change of the optical path difference. Therefore, when the ambient temperature changes, the length and effective index of the FBG will change. The optical path difference can be expressed as the following formula [46]:(1)$$\lambda_{b}^{}=2n_{eff}^{}\Lambda$$

Among them, Λ is the grating pitch and *n_eff_* is the effective refractive index of the core. When the ambient temperature acts on the FBG, the reflection spectrum will drift. In the formula, the coefficient of the thermal expansion and the optical path are constants. It is easy to obtain a reflection peak that is proportional to the temperature difference. In other words, the reflection peak changes with the outside temperature. We can obtain the ambient temperature by monitoring the frequency spectrum.
(2)$$\frac{d\lambda_{b}}{dT}=2\left(n_{eff}\frac{d\Lambda }{dT}+\Lambda \frac{dn_{eff}}{dT}\right)$$

The change of the grating pitch caused by the thermal expansion effect [47] is:(3)$$\frac{d\Lambda }{dT}=\alpha \cdot \Lambda$$

The change in the effective refractive index of the fiber caused by the thermo-optic effect is:(4)$$\frac{dn_{eff}}{dT}=n_{eff}\cdot \varepsilon$$

So the temperature sensitivity [48] is:(5)$$K_{T}=\frac{d\lambda_{b}}{dT}\cdot \frac{1}{\lambda_{b}}=\alpha +\varepsilon$$
(6)$$\alpha =\frac{d\Lambda }{dT}\cdot \frac{1}{\Lambda }$$
(7)$$\varepsilon =\frac{dn_{eff}}{dT}\cdot \frac{1}{n_{eff}}$$

In the formula, *K_T_* is the temperature sensitivity, α is the thermal expansion coefficient corresponding to the optical fiber material, and *ε* is the thermo-optic coefficient. Due to the difference in the doping composition and doping concentration, the expansion coefficient α and the thermo-optic coefficient *ε* of the various optical fibers are quite different, so the temperature response sensitivity will also be different.

In terms of the pressure, among all the external factors that cause the FBG wavelength shift, the most direct is the mechanical parameter. This is because no matter whether the grating is stretched or squeezed, it will cause the change of the grating period Λ, and the elasto-optical effect of the fiber itself makes the effective refractive index *n_eff_* also change with the change of the external stress state. Therefore, the use of FBG can be made into an optical fiber stress–strain sensor, where the wavelength shift caused by the stress can be uniformly described by Equation (8):(8)$$\Delta \lambda_{b}=2n_{eff}\Delta \Lambda +2\Delta n_{eff}\Lambda$$
where ΔΛ is the deformation of the fiber grid under stress, and Δ*n_eff_* is the elastic-optical effect of the fiber.

Differentiate both sides of the expression (1) of the center wavelength to obtain the following formula:(9)$$d\lambda_{b}=2n_{eff}d\Lambda +2\Lambda dn_{eff}$$

Divide both ends of Equation (9) by the terms on both sides of Equation (1) to obtain the following equation:(10)$$\frac{d\lambda_{b}}{\lambda_{b}}=\frac{dn_{eff}}{n_{eff}}+\frac{d\Lambda }{\Lambda }$$

Since the change of the refractive index of the optical fiber material is less affected by the stress, the influence of the refractive index, *n*, can be ignored. The above formula can be simplified to:(11)$$\frac{d\lambda_{b}}{\lambda_{b}}=\frac{d\Lambda }{\Lambda }=\frac{\Delta L}{L}$$

*L* represents the total length of the optical fiber, and Δ*L* represents the longitudinal expansion and contraction of the optical fiber.

In addition, since the grating is fixed on the elastic diaphragm, the change in the length of the grating Δ*L* is mainly affected by the elastic diaphragm. Therefore, the pressure response sensitivity of the two types of sensors is relatively similar from a theoretical point of view.

## 3. Experiment and Discussions

When conducting temperature experiments, keep the pressure constant and only change the temperature of the thermostat. The experimental temperature device is shown in Figure 3. The optical fiber grating demodulator used in this experiment adopts the U.S. MICRON OP ICS company SI 155; its minimum resolution is 0.02 nm, and the demodulator integrates a coupler, ASE light source (1510 nm–1590 nm), and signal demodulation system. The OMEGA thermometer has an accuracy of 0.5 °C. The length of the quartz fiber grating and the gold-plated grating in the experiment are both 3 mm, the wavelengths of the center after the structure is fixed are 1537.16 nm and 1551.48 nm, and the reflectivity is 90% and 30%, respectively.

In order to study the temperature response, the FBG is placed in a thermostat. The experiment carried out two temperature rising and falling experiments. The two temperature response curves can overlap well, and the four temperature response curves of the rising and falling temperatures can also overlap well, indicating that the sensor probe has good repeatability, as shown in Figure 4 and Figure 5. During the heating process, the center wavelength of the FBG decreases linearly with the increase in temperature. Similarly, during the cooling process, the center wavelength of the fiber grating decreases linearly with the decrease in temperature.

Figure 6 is a diagram of the pressure response of the FBG pressure sensor based on the data collected in Table 1, which is the temperature experimental data measured at 30 °C to 120 °C. The points in the figure are the measured data, and the fitting line is obtained by the least squares linear fitting. From the results of the experimental data analysis, the temperature response sensitivity of the sensor during the two heating processes are 0.009 nm/°C and 0.027 nm/°C, respectively. Combining the resolution of the spectrometer and the sensitivity of the gold-plated FBG, the sensor has a temperature resolution of 0.8 °C. Based on the measured wavelength, the sensitivity coefficient, and standard temperature value, the maximum temperature error is 2.5 °C, so the measurement accuracy δ can be obtained through the accuracy measurement equation as follows: δ = 100% − 2.5/(140 − 30) × 100%= 97.7%.

A pressure measurement system was designed, as shown in Figure 7. The probe was placed at the impulse tube of the pressure gauge. As the external pressure changes, the pressure diaphragm is deformed, and the grating is also deformed, which in turn causes the pitch of the grating to change. Therefore, the FBG pressure sensing system can demodulate the spectrum corresponding to the FBG at different pressure. The pressure source used in the experiment was the CW-600T pressure calibrator, which was connected with the sensor by the threaded interface (M20 × 1.5). The sensor, coupler, ASE light source, and demodulation system were connected to each other by optical fibers. The pressure gauge for monitoring the pressure source of the fiber grating pressure sensor was produced by the Xi’an Instrument Factory, with a range of 0–60 MPa and an accuracy of 0.01-level standard pressure gauge. The ASE light source emitted a beam of broadband light that entered the sensor probe through the coupler, and the modulation system demodulated the center wavelength of the FBG. The circular diaphragm was deformed due to the pressure, and the center wavelength at this time was less than the unpressurized center wavelength.

The experiment carried out two compression processes and monitored the peaks at 1537.04 nm and 1551.60 nm, respectively. During the boost process, the center wavelengths of the FBGs decreases linearly with the increase in the pressure, as shown in Figure 8 and Figure 9. Similarly, during the depressurization process, the center wavelengths of the FBGs increase linearly as the pressure decreases.

Table 2 is the experimental data collected under the different temperature environments, and Figure 10 is the FBG pressure response graph. The experimental results show that the pressure response sensitivity of the FBG and gold-plated FBG is similar, which is 0.3 nm/MPa at 0.1 MPa to 40 MPa. The sensor has good linearity, and the response curves of the two pressure rise-and-fall processes can be well overlapped. Combining the resolution of the spectrometer and the sensors’ sensitivity, the gold-plated FBG has a temperature resolution of 0.8 °C. Based on the measured wavelength, sensitivity coefficient, and standard pressure value, the maximum temperature error is 0.4 MPa, so the measurement accuracy can be obtained through the accuracy measurement equation as follows: δ = 100% − 0.4/(40 − 0.1) × 100% = 99.0%.

The response time of the temperature and pressure is a key parameter of the sensor, especially when the sensor is used in some extreme environments. Its influence mainly includes the following four parameters: the elastic deformation speed of the diaphragm structure, the change speed of the pressure to be measured, the change of the refractive index of the optical fiber material and the grid with the temperature and pressure, and the response time of the detector. A fast response experiment was carried out, and the response time of the FBG was measured by the time constant, which is defined as the time taken when the temperature or pressure rises to 63.2% of the steady-state value, that is, the collected signal rises from the initial value to 63.2%. The results show that the response times of the temperature and pressure are 2.7 ms/°C and 2 ms/MPa, respectively.

As shown in Figure 11, connect the sensor probe to the pressure gauge so that the sensor probe can sense the pressure signal, and at the same time place the sensor in the thermostat, so that the sensor can sense temperature and pressure information at the same time. Based on the dual-parameter sensor probe, the temperature and pressure are measured at the same time, and the spectral data at different moments are obtained, as shown in Figure 12.

When the temperature and pressure act on the sensor, the sensitivity matrix equation can be used to achieve the dual-parameter differential measurement, such as Equation (12), where *k_x_* is the temperature and pressure response sensitivity of the two sensors, *λ*^0^ is the initial wavelength, T is the temperature to be measured in the experiment, and the subscripts 1 and 2 are used to distinguish two different sensor structures. The wavelength *λ* can be expressed as follows:(12)$$\left[\begin{array}{c} \lambda_{1}\\ \lambda_{2} \end{array}\right]=\left[\begin{array}{c} \lambda_{1}^{0}\\ \lambda_{2}^{0} \end{array}\right]+\left[\begin{array}{cc} k_{1} & k_{2}\\ k_{3} & k_{4} \end{array}\right]\left[\begin{array}{c} T\\ P \end{array}\right]$$
(13)$$\left[\begin{array}{c} \lambda_{1}-\lambda_{1}^{0}\\ \lambda_{2}-\lambda_{2}^{0} \end{array}\right]=\left[\begin{array}{cc} k_{1} & k_{2}\\ k_{3} & k_{4} \end{array}\right]\left[\begin{array}{c} T\\ P \end{array}\right]$$

By multiplying the reciprocal matrix and combining the formula, the temperature and pressure parameters can be obtained as follows:(14)$${\left[\begin{array}{cc} k_{T1} & k_{P}\\ k_{T2} & k_{P} \end{array}\right]}^{-1}\left[\begin{array}{c} \lambda_{1}-\lambda_{1}^{0}\\ \lambda_{2}-\lambda_{2}^{0} \end{array}\right]={\left[\begin{array}{cc} k_{T1} & k_{P}\\ k_{T2} & k_{P} \end{array}\right]}^{-1}\left[\begin{array}{cc} k_{T1} & k_{P}\\ k_{T2} & k_{P} \end{array}\right]\left[\begin{array}{c} T\\ P \end{array}\right]$$
(15)$$\left[\begin{array}{c} T\\ P \end{array}\right]={\left[\begin{array}{cc} k_{T1} & k_{P}\\ k_{T2} & k_{P} \end{array}\right]}^{-1}\left[\begin{array}{c} \lambda_{1}-\lambda_{1}^{0}\\ \lambda_{2}-\lambda_{2}^{0} \end{array}\right]$$

The initial wavelengths $\lambda_{1}^{0}$ and $\lambda_{2}^{0}$ of the detection wavelengths in the experiment are 1537.04 nm and 1551.60 nm, respectively. The temperature response sensitivities, *k_Tx_*, of the two sensing structures are 0.008 nm/°C and 0.024 nm/°C, respectively, and the pressure response sensitivities *k_Px_* are both 0.3 nm/MPa. Substituting each parameter into formula (15) can obtain formula (16), which is used to measure the environmental parameters. Specifically, with the help of the matrix method, the temperature and pressure values at different time points can be detected at the same time, as shown in Figure 13.
(16)$$\left[\begin{array}{c} T\\ P \end{array}\right]={\left[\begin{array}{cc} 0.008 & 0.3\\ 0.024 & 0.3 \end{array}\right]}^{-1}\left[\begin{array}{c} \lambda_{1}-1537.04\\ \lambda_{2}-1551.60 \end{array}\right]$$

## 4. Conclusions

This paper proposes and manufactures an all-optical fiber sensor system based on the parallel structure of gold-plated FBG and quartz FBG, which can simultaneously measure temperature and pressure. As the temperature and pressure sensitivity of the two sensor structures are different, we can measure the temperature and pressure by monitoring the response of the wavelength in real-time based on the sensitivity matrix equation. The experimental results show that the pressure response sensitivity of the quartz FBG and gold-plated FBG are both 0.3 nm/MPa. The temperature sensitivity of the gold-plated FBG is 0.024 nm/°C with a resolution of 0.8 °C, and the sensitivity of the quartz FBG is 0.008 nm/°C with a resolution of 0.067 MPa, which can distinguish the temperature and pressure changes well. The sensor probe has the advantages of a simple structure, easy production, small size, high sensitivity, and dual-parameter measurement, which can be applied to monitor the running status of the EHA.

## Figures and Tables

**Figure 1 micromachines-13-00195-f001:**
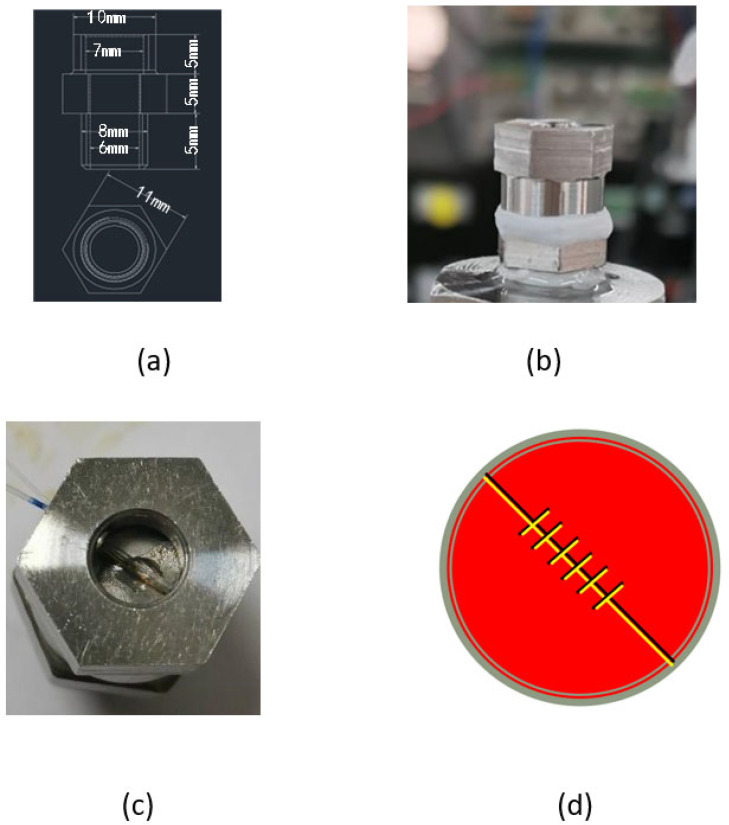
Design and manufacture of the sensor probe. (**a**) Structural design drawing of the sensor probe; (**b**) Physical map of the sensor probe; (**c**) Hexagonal nut; (**d**) Schematic diagram of the installation of parallel grating sensors.

**Figure 2 micromachines-13-00195-f002:**
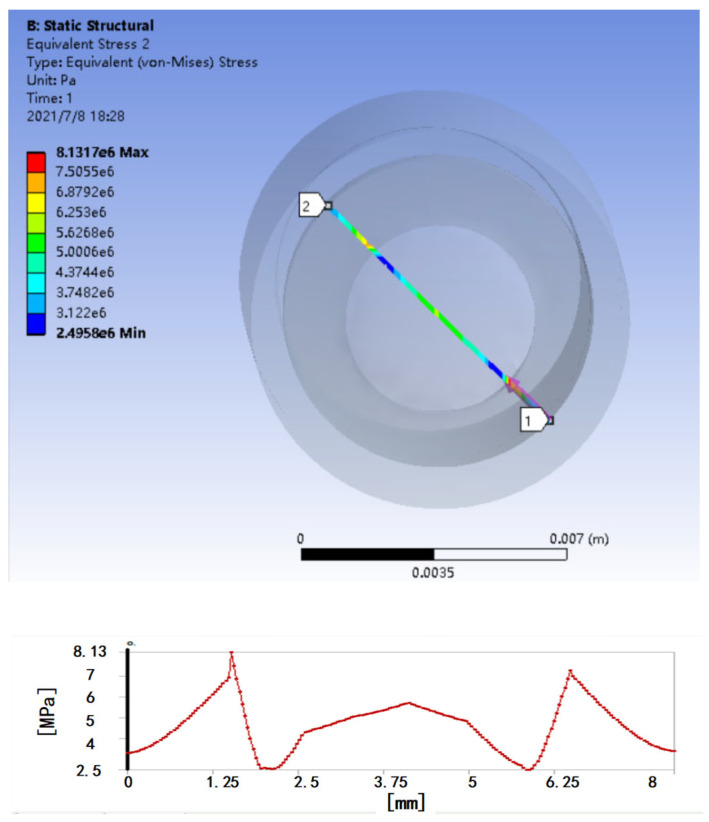
Force analysis of the pressure diaphragm.

**Figure 3 micromachines-13-00195-f003:**
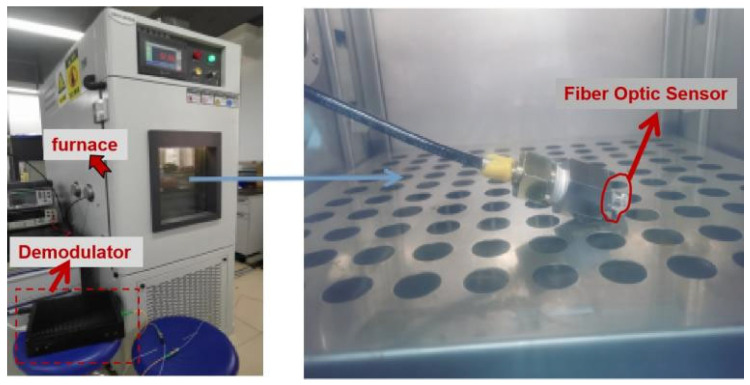
The actual photograph for the temperature measurement system.

**Figure 4 micromachines-13-00195-f004:**
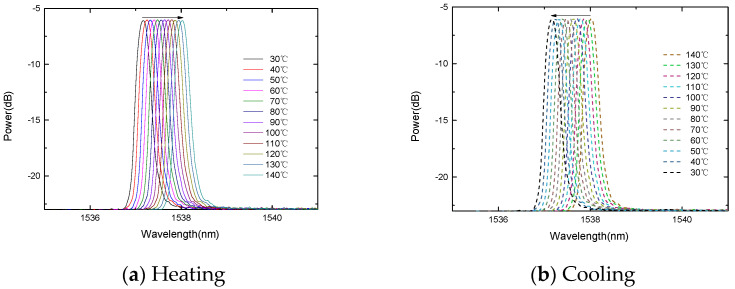
The change of the FBG center wavelength with the temperature.

**Figure 5 micromachines-13-00195-f005:**
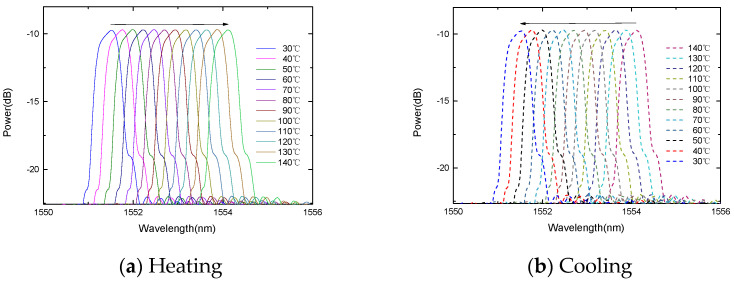
The change of the gold-plated FBG center wavelength with the temperature.

**Figure 6 micromachines-13-00195-f006:**
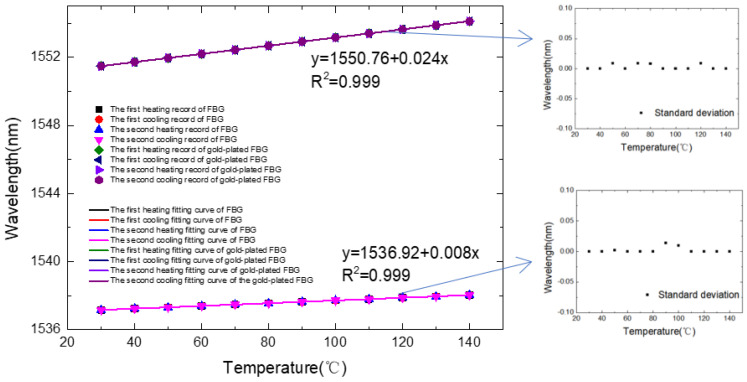
Fitting curve of the temperature response sensitivity of the double grating sensor probe and the error bars.

**Figure 7 micromachines-13-00195-f007:**
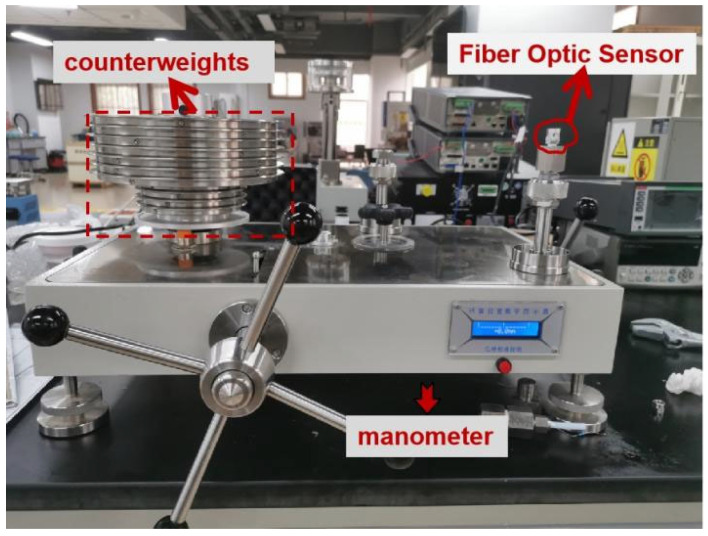
The actual photograph for the pressure measurement system.

**Figure 8 micromachines-13-00195-f008:**
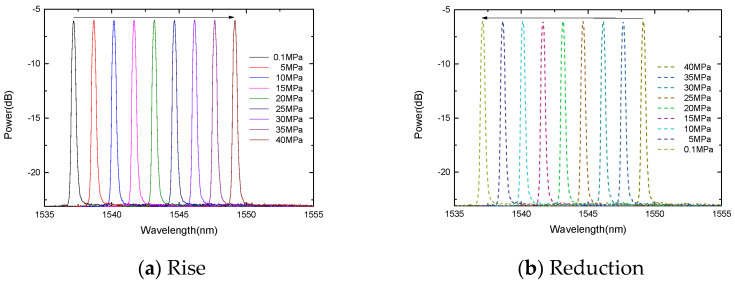
The change of the FBG center wavelength with pressure.

**Figure 9 micromachines-13-00195-f009:**
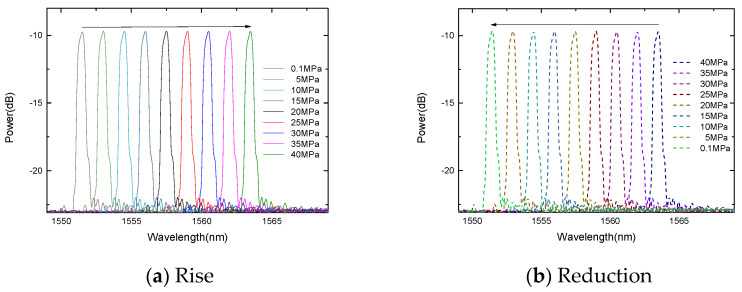
The change of the gold-plated FBG center wavelength with pressure.

**Figure 10 micromachines-13-00195-f010:**
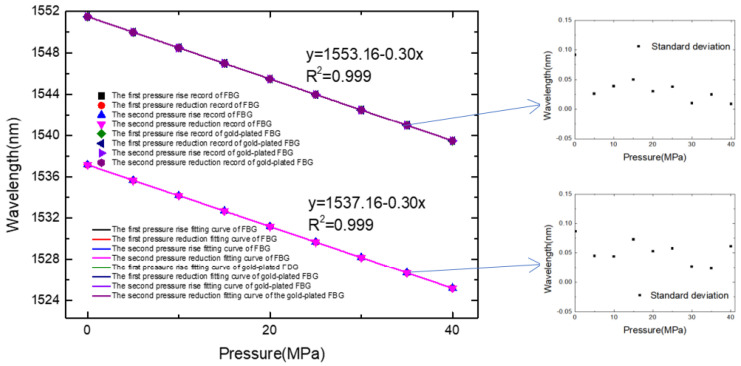
Fitting curve of the pressure response sensitivity of the double grating sensor probe and the error bars.

**Figure 11 micromachines-13-00195-f011:**
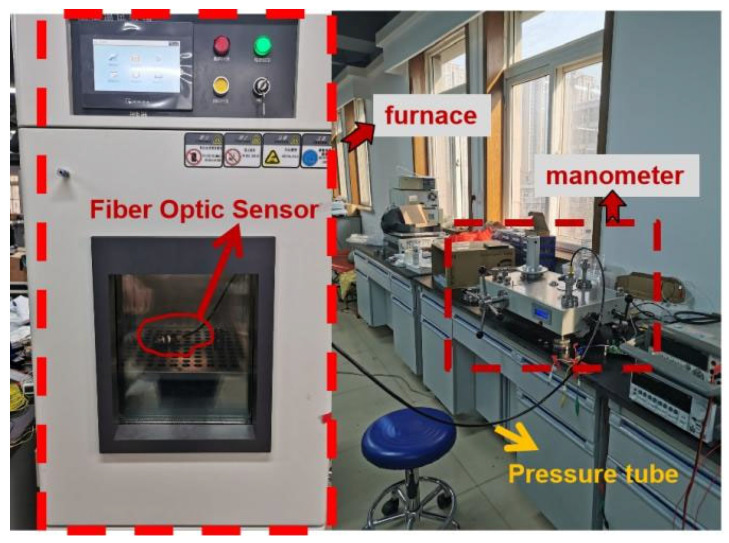
Dual-parameter sensing experiment platform.

**Figure 12 micromachines-13-00195-f012:**
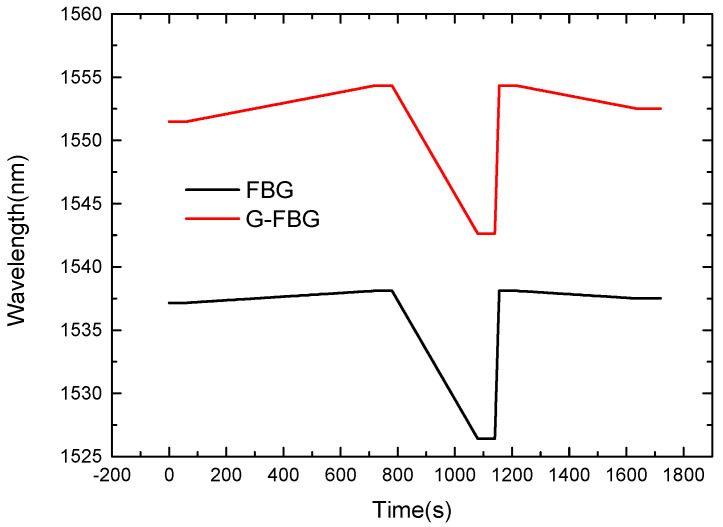
Measure the spectra with temperature and pressure.

**Figure 13 micromachines-13-00195-f013:**
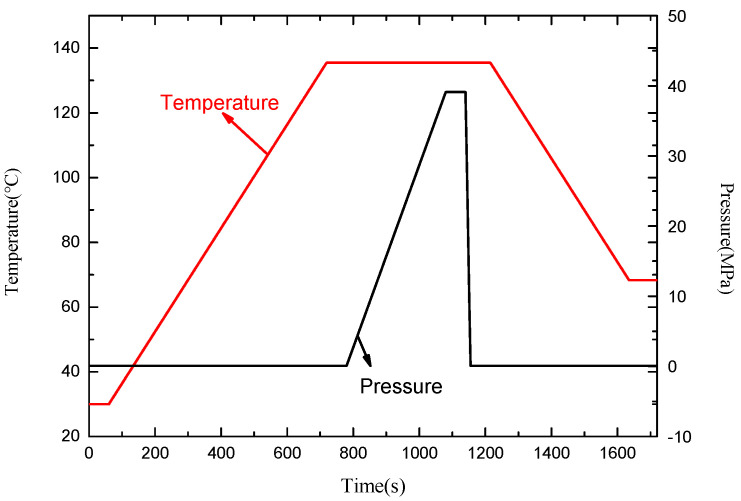
The temperature and pressure conditions to be measured based on the spectrum analysis of the sensor probe.

**Table 1 micromachines-13-00195-t001:** Wavelength data collected during the temperature rise and fall.

	T/°C	Wavelength Corresponding to the First Temperature Change of the FBG/nm	Wavelength Corresponding to the Second Temperature Change of the FBG/nm	Wavelength Corresponding to the First Temperature Change of the Gold-Plated FBG/nm	Wavelength Corresponding to the Second Temperature Change of the Gold-Plated FBG/nm
↑	30	1537.16	1537.16	1551.48	1551.48
	40	1537.24	1537.24	1551.72	1551.72
	50	1537.32	1537.32	1551.96	1551.96
	60	1537.4	1537.4	1552.2	1552.2
	70	1537.48	1537.46	1552.44	1552.44
	80	1537.56	1537.56	1552.68	1552.68
	90	1537.64	1537.64	1552.94	1552.90
	100	1537.72	1537.72	1553.16	1553.16
	110	1537.8	1537.8	1553.4	1553.4
	120	1537.86	1537.88	1553.64	1553.64
	130	1537.96	1537.96	1553.88	1553.88
	140	1538.04	1538.04	1554.12	1554.12
↓	140	1538.04	1538.04	1554.12	1554.12
	130	1537.96	1537.96	1553.88	1553.88
	120	1537.88	1537.88	1553.64	1553.64
	110	1537.8	1537.8	1553.4	1553.4
	100	1537.72	1537.72	1553.16	1553.18
	90	1537.64	1537.64	1552.92	1552.92
	80	1537.54	1537.56	1552.68	1552.68
	70	1537.48	1537.48	1552.44	1552.44
	60	1537.4	1537.4	1552.2	1552.2
	50	1537.32	1537.34	1551.96	1551.96
	40	1537.24	1537.24	1551.72	1551.72
	30	1537.16	1537.16	1551.48	1551.48

**Table 2 micromachines-13-00195-t002:** Wavelength data collected during the pressure rise and fall.

	P/MPa	Wavelength Corresponding to the First Pressure Change of the FBG/nm	Wavelength Corresponding to the Second Pressure Change of the FBG/nm	Wavelength Corresponding to the First Pressure Change of the Gold-Plated FBG/nm	Wavelength Corresponding to the Second Pressure Change of the Gold-Plated FBG/nm
↑	0.1	1537.24	1537.04	1551.6	1551.5
	5	1535.66	1535.66	1549.96	1550.08
	10	1534.08	1534.18	1548.54	1548.48
	15	1532.68	1532.78	1546.98	1546.92
	20	1531.18	1531.18	1545.48	1545.49
	25	1529.73	1529.66	1544	1543.98
	30	1528.14	1528.14	1542.58	1542.54
	35	1526.66	1526.68	1541	1540.99
	40	1525.16	1525.18	1539.54	1539.38
↓	40	1525.16	1525.16	1539.5	1539.48
	35	1526.66	1526.72	1540.98	1541.04
	30	1528.16	1528.16	1542.6	1542.58
	25	1529.66	1529.74	1544.06	1544.08
	20	1531.24	1531.16	1545.48	1545.38
	15	1532.66	1532.66	1547	1547.08
	10	1534.16	1534.16	1548.54	1548.44
	5	1535.72	1535.66	1550.04	1550.04
	0.1	1537.16	1537.28	1551.58	1551.38

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
