# Peer review of "Research on the High Temperature and High Pressure Gold-Plated Fiber Grating Dual-Parameter Sensing Measurement System"

_micromachines, 2022, doi:10.3390/mi13020195_

Round 1

Reviewer 1 Report

In this paper, the authors introduced a gold-plated fiber grating for simultaneous detection of high temperature and pressure is proposed and analyzed. The idea behind the work is interesting. However, I still have quite a number of concerns in this manuscript. Please see find the comments below.

 1. The information for the sensor structure (figure 1) is not enough. The authors should give much more information about the fabrication and installation steps. So, the readers can get its reproducibility. 

 2. The authors should give much more information about the novelty of this paper, especially the effect of using this gold grating, are they really useful for sensing high temperature and pressure?

3. Authors are suggested to discuss the reason for using gold.

4. Please mention the resolution?

5. The spectra 4, 5, 8, 9 are not clear from the reader's point of view.

6. The English require a careful check.

Author Response

Dear editor and reviewer:

    Thank you very much for your comments about our paper submitted to Research on high temperature high pressure gold-plated fiber grating dual-parameter sensing measurement system (micromachines-1562642). We have learned much from your comments, which are fair, encouraging and constructive. After carefully studying the comments and your advice, we have made corresponding changes. And we also add some experiments in this manuscript. 

The following is the answers and revisions we have made in response to the reviewer's questions and suggestions item by item.

  1. Comments:In this paper, the authors introduced a gold-plated fiber grating for simultaneous detection of high temperature and pressure is proposed and analyzed. The idea behind the work is interesting. However, I still have quite a number of concerns in this manuscript. Please see find the comments below.

The information for the sensor structure (figure 1) is not enough. The authors should give much more information about the fabrication and installation steps. So, the readers can get its reproducibility.

   Revision: Thanks for your suggestion, we have supplemented the information in the sensing structure section of Figure 1. The font in the figure has been enlarged to make the structure clearer. We have introduced that the probe is divided into a hexagonal nut and a bottom-end connector, and the diaphragm is held between the two structures by assembly, and a sentence describing the probe has been added to the manuscript.

   The location of the revision: Figure1ï¼›On lines 165-170.

  1. Comments:The authors should give much more information about the novelty of this paper, especially the effect of using this gold grating, are they really useful for sensing high temperature and pressure?

   Revision: Thanks for your prompts, we have reviewed related papers, studied the current state of development of gold-plated gratings, and added the latest research papers in this field to the manuscript. It is found that the gold-coated grating has certain advantages in multi-parameter sensing and high temperature sensitivity sensor, which is the research direction of the grating in the future.

On the one hand, there are few studies on gold plating, and it has been found that gold-plated sensors respond to temperature and strain through existing literature [1-5]. Combined with the project requirements, by designing the diaphragm structure and using the temperature and strain characteristics of the gold-plated grating, we carried out temperature and pressure sensing. On the other hand, our previous research has confirmed that the temperature measurement of the pure quartz grating we developed can reach 800℃[6], and it can still withstand high temperature after gold plating, and high-sensitivity optical fiber sensing can be realized by using the thermal expansion characteristics of metal[3-5]. And then through the double grating structure, the temperature and pressure dual parameter sensing is realized. We have added responsive segments and references to the manuscript.

  • Monaghan T, Capel A J, Christie S D, et al. Solid-state additive manufacturing for metallized optical fiber integration[J]. Composites Part A: Applied Science and Manufacturing, 2015, 76: 181-193.
  • Yanchao L, Jin F, Chong X, et al. The feasibility of gold-plated fiber Bragg grating sensors for in-situ detection of lithium-ion batteries[J]. Progress in Laser and Optoelectronics, 2017, 54(4): 105-111.
  • Laarossi I, Roldán-Varona P, Quintela-Incera M A, et al. Ultrahigh temperature and strain hybrid integrated sensor system based on Raman and femtosecond FBG inscription in a multimode gold-coated fiber[J]. Optics express, 2019, 27(26): 37122-37130.
  • Wu H, Lin Q, Jiang Z, et al. A temperature and strain sensor based on a cascade of double fiber Bragg grating[J]. Measurement Science and Technology, 2019, 30(6): 065104.
  • Yanjun Z, Haichuan G, Longtu Z, et al. Embedded gold-plated fiber Bragg grating temperature and stress sensors encapsulated in capillary copper tube[J]. Opto-Electronic Engineering, 2021, 48(3): 200195-1-200195-11.
  • Zhao N, Lin Q, Yao K, et al. Simultaneous Measurement of Temperature and Refractive Index Using High Temperature Resistant Pure Quartz Grating Based on Femtosecond Laser and HF Etching[J]. Materials, 2021, 14(4): 1028.

   The location of the revision: On lines 77-79,88-91,128-131; reference 27,45.

  1. Comments:Authors are suggested to discuss the reason for using gold.

   Revision: Thanks for your comment, combined with the reply to the second comment, we mainly summarize the reasons for choosing gold-plated gratings as follows:

First, compared with quartz gratings, gold-coated gratings have higher sensitivity to temperature response, which has research value;

Secondly, the gold-plated grating is responsive to temperature and strain. By designing a composite sensing structure, multi-parameter sensing can be realized to complete the project requirements;

Finally, the research on FBG is relatively mature. Combined with our previous research, pure quartz FBG can withstand up to 800 ℃, while gold is a high-temperature resistant metal, which can also withstand high temperature when coated on grating, realizing high-sensitivity high-temperature sensor. 

   The location of the revision: On lines 77-79.

  1. Comments:Please mention the resolution?

   Revision: We have added resolution to both the experimental and conclusion sections. The optical fiber grating demodulator used in this experiment adopts the resolution of 0.02 nm, and the OMEGA thermometer accuracy is 0.5℃. The fiber optic sensor developed in the manuscript has a pressure resolution of 0.067 MPa and a temperature resolution of 0.8℃.

   The location of the revision: The fourth sentence of the paragraph above Figure 6; the fourth sentence of the paragraph above Figure 10; the third and fifth sentences of the paragraph above Figure 4; the fourth sentence of the conclusions section.

  1. Comments: The spectra 4, 5, 8, 9 are not clear from the reader's point of view.

   Revision: Thanks for your suggestion. Considering the unclear problem of spectra 4, 5, 8, 9, we have separated the spectra of the processes of heating and cooling, pressure rise and fall, so that readers can clearly see the changes of the spectrum with the external physical quantities.

   The location of the revision: The Figure 4, 5, 8, 9.   

  1. Comments:The English require a careful check.

   Revision: Thanks to the reviewer's suggestion, we asked an English teacher to help us check the manuscript. The Abstract, Introduction, Experiments, and Conclusions sections  have been revised and checked, and the revised sections have been highlighted in the manuscript. Thanks again to the reviewer for the suggestions on various aspects of the manuscript so that our manuscript can be better presented to readers.

   The location of the revision: On lines 27-30, 35, 70, 297-298, 378-381.

   With best regards!

Yours faithfully

Reviewer 2 Report

The paper reports the research on high-temperature high-pressure gold-plated fiber grating dual-parameter sensing measurement system for electrohydrostatic drive actuator. The results shown in this manuscript are indeed impressive. This paper is possibly publishable but should be revised again. For improving a manuscript, it is advisable to address the following comments:

  1. The title of this paper indicates that the application of this new sensing system is specifically in EHA. In the experiment section, the tests in EHA should be included and explored. Otherwise, it recommends revising the application scenario in the title.
  2. In the introduction section, please point out what your sensing range of the high pressure and high temperature is. Moreover, please explain why the current techniques are not able to perform well in those ranges.
  3. In Figure 1(a), please increase the font size of the dimensions of the sensing probe and label the unit.
  4. In Figure 1(d), it looks that the grating is not perpendicular to the fiber, is it the tilted FBG, and what is the tilted angel?
  5. On lines 151-153, what’s the grating pitch the FBG used?
  6. If the equations in the manuscript are not original, please provide citations for them.
  7. On lines 250-252, I think the accuracy is 97.7% (100%-2.3%). The accuracy formula provides accuracy as a difference of error rate from 100%. To find accuracy it first needs to calculate the error rate. And the error rate is the percentage value of the difference of the observed and the actual value, divided by the actual value.
  8. On the line of 288, I think the accuracy should be 99% (100%-1%).
  9. Please check the typo and grammar in the entire manuscript.

Author Response

Dear editor and reviewer:

    Thank you very much for your comments about our paper submitted to Research on high temperature high pressure gold-plated fiber grating dual-parameter sensing measurement system (micromachines-1562642). We have learned much from your comments, which are fair, encouraging and constructive. After carefully studying the comments and your advice, we have made corresponding changes. And we also add some experiments in this manuscript. 

The following is the answers and revisions we have made in response to the reviewer's questions and suggestions item by item.

  1. Comments: The paper reports the research on high-temperature high-pressure gold-plated fiber grating dual-parameter sensing measurement system for electrohydrostatic drive actuator. The results shown in this manuscript are indeed impressive. This paper is possibly publishable but should be revised again. For improving a manuscript, it is advisable to address the following comments:

The title of this paper indicates that the application of this new sensing system is specifically in EHA. In the experiment section, the tests in EHA should be included and explored. Otherwise, it recommends revising the application scenario in the title.

    Revision: Thanks to the reviewer's suggestion, due to the limitation of the project's confidentiality requirements, this field experiment of the sensing system specifically applied in the EHA cannot be directly presented in the manuscript. Combined with the content and structure of the manuscript, as well as the reviewer's suggestion, we have revised the title "Research on high temperature high pressure gold-plated fiber grating dual-parameter sensing measurement system for electrohydrostatic drive actuator" to "Research on high temperature high pressure gold-plated fiber grating dual-parameter sensing measurement system", and have deleted the content of the inappropriate application scenarios.

   The location of the revision: Title. 

  1. Comments:In the introduction section, please point out what your sensing range of the high pressure and high temperature is. Moreover, please explain why the current techniques are not able to perform well in those ranges.

   Revision: In terms of pressure measurement, the pressure range of pure optical fiber structures is generally in the order of kPa[1-3]. When adding cantilever beams, diaphragms and other structures, it can reach the level of MPa[5-9]. Combined with the working conditions required by the project, we choose the diaphragm structure, and the measurement pressure can reach 60MPa, which belongs to the high pressure range.

In terms of temperature measurement, for individual temperature measurement, the commonly used quartz FBG can reach 400℃, and the gold-plated pure quartz FBG used in our project can even reach 800℃[10]. However, for temperature measurement under high pressure, current research generally stays at room temperature, and 80℃ is considered a relatively high temperature [11].

The manuscript describes the simultaneous measurement of temperature and pressure. In order to meet the needs of actual working conditions, the temperature measurement can reach 120℃, and the pressure can reach 60MPa, which can meet the sensing measurement requirements of EHA. Combined with the reviewer's comments, and in order to better describe the sensing range of high temperature and high pressure, we have added relevant description in the introduction section, and marked them with highlights.

[1] Schenato L, Aneesh R, Palmieri L, et al. Fiber optic sensor for hydrostatic pressure and temperature measurement in riverbanks monitoring[J]. Optics & Laser Technology, 2016, 82: 57-62.

[2] Liu Y, Jing Z, Li R, et al. Miniature fiber-optic tip pressure sensor assembled by hydroxide catalysis bonding technology[J]. Optics express, 2020, 28(2): 948-958.

[3] Cheng X, Dash J N, Gunawardena D S, et al. Silicone Rubber Based Highly Sensitive Fiber-Optic Fabry–Perot Interferometric Gas Pressure Sensor[J]. Sensors, 2020, 20(17): 4927.

[4] Liu Y, Jing Z, Liu Q, et al. Differential-pressure fiber-optic airflow sensor for wind tunnel testing[J]. Optics Express, 2020, 28(17): 25101-25113.

[5] Lijun L, Xu Z, Bin T, et al. A miniature fiber grating rock pressure sensor[J]. Journal of China Coal Society, 2013, 38(11): 2084-2088. Doi:10.13544/j.cnki.jeg.2015.06.008

[6] Hui W, Yang Y, Bing L. Demodulation method of fiber grating pressure sensor based on dense wavelength division multiplexer[J]. Progress in Laser and Optoelectronics, 2016, 53(4): 220-226.

[7] Zhang H, Song W, Wang Z, et al. Numerical and experimental studies of high-sensitivity plug-in pressure sensor based on fiber Bragg gratings[J]. Optical Engineering, 2016, 55(9): 096104.

[8] Wang Y, Wang M, Xia W, et al. Optical fiber Bragg grating pressure sensor based on dual-frequency optoelectronic oscillator[J]. IEEE Photonics Technology Letters, 2017, 29(21): 1864-1867.

[9] Fan Q, Feng D, Yu D, et al. Design and investigation of the fiber Bragg grating pressure sensor based on square diaphragm and truss-beam structure[J]. Optical Engineering, 2019, 58(9): 097109.

[10]Zhao N, Lin Q, Yao K, et al. Simultaneous Measurement of Temperature and Refractive Index Using High Temperature Resistant Pure Quartz Grating Based on Femtosecond Laser and HF Etching[J]. Materials, 2021, 14(4): 1028.

[11] Yanjun Z, Haichuan G, Longtu Z, et al. Embedded gold-plated fiber Bragg grating temperature and stress sensors encapsulated in capillary copper tube[J]. Opto-Electronic Engineering, 2021, 48(3): 200195-1-200195-11.

   The location of the revision: The second and penultimate sentences of the fourth paragraph of the introduction section; references 33-41; references 45 .

  1. Comments:In Figure 1(a), please increase the font size of the dimensions of the sensing probe and label the unit.

   Revision: We have enlarged the dimension annotation of the sensing probe and marked the units in Figure 1(a). The overall diameter of the probe is 11mm and the height is 5mm. A specific description of the probe has been added to the manuscript.

   The location of the revision: Figure1(a), on lines 165-170.   

  1. Comments:In Figure 1(d), it looks that the grating is not perpendicular to the fiber, is it the tilted FBG, and what is the tilted angel? 

   Revision: Thank you for helping to point this out, we're very sorry, the drawing standards are now standardized. Since we are using a vertical grating, we have changed the grating on the diagram to a vertical structure to prevent misunderstandings.

   The location of the revision: Figure 1(d).

  1. Comments:On lines 151-153, what’s the grating pitch the FBG used?

   Revision: Thank you very much for the reminder, we have included the corresponding description of the grating pitch in the manuscript and highlighted it. The gold-coated FBG and reference quartz FBG used in the experiments have different grating pitches of 523.52 nm and 528.40 nm, respectively . This can also be given by the fiber grating equation, as in equation 1.

                             (1)

The refractive index of the pure silica fiber core used in the experiment is 1.4681. As shown in equaiton 2, when the center wavelengths are 1537.16 nm and 1551.48 nm, the grating pitches are 523.52 nm and 528.40 nm, respectively.

                           (2)

   The location of the revision: On lines 175-179.   

  1. Comments:If the equations in the manuscript are not original, please provide citations for them.

   Revision: Thank you for your suggestion. We have provided references for the basic equations that are not original in the manuscript for the definition and interpretation of the basic equations.

   The location of the revision: Reference46-48.

  1. Comments:On lines 250-252, I think the accuracy is 97.7% (100%-2.3%). The accuracy formula provides accuracy as a difference of error rate from 100%. To find accuracy it first needs to calculate the error rate. And the error rate is the percentage value of the difference of the observed and the actual value, divided by the actual value.

   Revision: Thanks for your suggestion, and we have checked some literature and found that the formula"δ=ΔT/(Tmax-Tmin)×100%" is used to measure error rate rather than accuracy. We have replaced "2.3%" with "97.7%" in the manuscript.

   The location of the revision: On lines 288.

  1. Comments:On the line of 288, I think the accuracy should be 99% (100%-1%).

   Revision: Thanks for your reminder, we have changed the accuracy to 99%.

   The location of the revision: On lines 330.

  1. Comments:Please check the typo and grammar in the entire manuscript.9 请检查整篇稿件中的错字和语法。

   Revision: Thank you for your suggestion. We have checked the manuscrip with the help of English teacherst. The inappropriate grammar and the typo have been identified and revised. Finally, the revised section has been highlighted in the manuscript.

   The location of the revision: On lines 27-30, 35, 297-298, 378-381, 387.

   With best regards!

Yours faithfully

Reviewer 3 Report

This paper describes the development of a high temperature high pressure gold-plated fiber grating dual-parameter sensing measurement system for electrohydrostatic drive actuator. This is a very interesting work in which the authors manage to perform simulataneus measurements of pressure and temperature with this convenient optical fiber structure. As a result, the temperature sensitivity of the gold-plated FBG is 3 times that of quartz FBG, so the system can be used for monitoring complex environments with high temperature and high pressure. The paper is well written, although the English language could be slightly improved. The Introduction provides a good background to the reader and the basic references to understand the context of the work.  The manuscript includes high quality figures properly illustrating the key concepts of the work. The results are concise and potentially applicable in practical scenarios. Finally, the Conclusions are well supported by the obtained data. All in all, this is a very good work and an excellent match for Micromachines. I will just include a set of minor but mandatory revisions to be performed before the paper is considered for publication:

 - In the Introduction, the authors write "FBG has attracted much attention due to its small size, low cost, easy manufacturing and high maturity". Then, they start referencing some works starting from 2003, which is almost 20 years ago. I recommend to replace this examples by more recent ones. You can include this work [Sensors and Actuators A: Physical 332, 113061 (2021)] as an example of FBG integration in tiny locations and this other work [Photonics 8(10), 450 (2021)] as an example of high maturity, among others.

 - The authors describe the design of the sensing structure in Section 2. They mention that they use ultraviolet light to write the grating structures. However, they do not provide additional details. Do the authors photo-inscribe the FBGs using the well-known phase-mask technique, they use point-by-point photo-inscription or do they use another technique?

 - In Section 3 the authors say that the FBGs have a reflectivity of 90%. However, there is an optical power difference between the FBG and the gold-plated FBG, as it can be seen when comparing Fig. 4 vs Fig 5 or Fig 8 vs Fig 9. It can be seen that the FBGs have a difference of roughly 5dB. Why does this happen?

 - Could the authors insert the error bars in Figures 6 and 10?

Author Response

Dear editor and reviewer:

    Thank you very much for your comments about our paper submitted to Research on high temperature high pressure gold-plated fiber grating dual-parameter sensing measurement system (micromachines-1562642). We have learned much from your comments, which are fair, encouraging and constructive. After carefully studying the comments and your advice, we have made corresponding changes. And we also add some experiments in this manuscript. 

The following is the answers and revisions we have made in response to the reviewer's questions and suggestions item by item.

  1. Comments:This paper describes the development of a high temperature high pressure gold-plated fiber grating dual-parameter sensing measurement system for electrohydrostatic drive actuator. This is a very interesting work in which the authors manage to perform simulataneus measurements of pressure and temperature with this convenient optical fiber structure. As a result, the temperature sensitivity of the gold-plated FBG is 3 times that of quartz FBG, so the system can be used for monitoring complex environments with high temperature and high pressure. The paper is well written, although the English language could be slightly improved. The Introduction provides a good background to the reader and the basic references to understand the context of the work.  The manuscript includes high quality figures properly illustrating the key concepts of the work. The results are concise and potentially applicable in practical scenarios. Finally, the Conclusions are well supported by the obtained data. All in all, this is a very good work and an excellent match for Micromachines. I will just include a set of minor but mandatory revisions to be performed before the paper is considered for publication:

In the Introduction, the authors write "FBG has attracted much attention due to its small size, low cost, easy manufacturing and high maturity". Then, they start referencing some works starting from 2003, which is almost 20 years ago. I recommend to replace this examples by more recent ones. You can include this work [Sensors and Actuators A: Physical 332, 113061 (2021)] as an example of FBG integration in tiny locations and this other work [Photonics 8(10), 450 (2021)] as an example of high maturity, among others.

   Revision: Thanks to the reviewer's suggestion, we have included some recent work in the manuscript. Includes introduction and description of references [Sensors and Actuators A: Physical 332, 113061 (2021)], [Photonics 8(10), 450 (2021)]. Corresponding changes have been made in the manuscript and are highlighted.

   The location of the revision: On lines 65-70; reference 21,22.

  1. Comments:The authors describe the design of the sensing structure in Section 2. They mention that they use ultraviolet light to write the grating structures. However, they do not provide additional details. Do the authors photo-inscribe the FBGs using the well-known phase-mask technique, they use point-by-point photo-inscription or do they use another technique?

   Revision: Thank you for your comment. We use UV mask technology for marking. Excimer laser is used to irradiate the mask. The UV light passes through the mask to form diffraction fringes and irradiates the single-mode fiber. Finally, a grating area with uniform refractive index changes is formed in the core of the fiber. The refractive index of the pure silica fiber core used in the experiment is 1.4681. And the gold-coated FBG and reference quartz FBG used in the experiments have different grating pitches of 523.52 nm and 528.40 nm, and the central wavelengths finally obtained are 1537.16 nm and 1551.48 nm, respectively. We have included the corresponding content in the manuscript and highlighted it.

   The location of the revision:  On lines 175-179.  

  1. Comments:In Section 3 the authors say that the FBGs have a reflectivity of 90%. However, there is an optical power difference between the FBG and the gold-plated FBG, as it can be seen when comparing Fig. 4 vs Fig 5 or Fig 8 vs Fig 9. It can be seen that the FBGs have a difference of roughly 5dB. Why does this happen?

   Revision: Thank you for your question and we are very sorry. When making gold-plated gratings, FBGs with a reflectivity of 30% and 90% are used. Finally, the FBG with 90% reflectivity and the gold-plated FBG with 30% reflectivity are presented in the manuscript, and there is a 5dB difference between them. At the same time, we have produced 90% gold-plated FBGs with similar experimental results to 30% gold-plated FBGs. Therefore, the reflectivity has little effect on the experimental results of temperature and pressure measurement. We have revised the description of grating reflectance in the manuscript. Thanks again to the reviewers for raising this issue and making our manuscript more complete.

   The location of the revision: The last sentence of the paragraph above in Figure 4.

  1. Comments:Could the authors insert the error bars in Figures 6 and 10?

   Revision: Thanks to the reviewer for the suggestion, we also felt the need to add error analysis to the figure. Since there are already many annotations in Fig. 6 and Fig. 10, in order to present the experimental results more clearly, we have added small graphs characterizing the error situation to Fig. 6 and Fig. 10 to improve the description of the experiment. Thank you again for your contribution to improve the quality of our manuscripts.

   The location of the revision: Figure 6, 10.

   With best regards!

Yours faithfully

Round 2

Reviewer 1 Report

The authors did the requested amendments. Hence I recommend the manuscript for its publication in the current form.

Reviewer 2 Report

The authors have revised the manuscript carefully according to the requisitions and tried their best to resolve the problems in the comments. Therefore, I think that this revised manuscript should be accepted and published in this journal.